# Fostering Tolerance and Respect for Diversity through the Fundamentals of Islamic Education

Semiyu Adejare Aderibigbe [1,2,*], Mesut Idriz [1], Khadeegha Alzouebi [3], Hussain AlOthman [1], Wafa Barhoumi Hamdi [1] and Assad Asil Companioni [1]

[1] College of Arts, Humanities and Social Sciences, University of Sharjah,
Sharjah P.O. Box 27272, United Arab Emirates
[2] Institute of Leadership in Higher Education, University of Sharjah, Sharjah P.O. Box 27272, United Arab Emirates
[3] School of e-Education, Hamdan Bin Mohamed Smart University, Dubai P.O. Box 71400, United Arab Emirates
[*] Correspondence: saderibigbe@sharjah.ac.ae

**Abstract:** Societies are getting more diverse, with social issues increasing, necessitating the need to intensify efforts to promote tolerance and respect for diversity. In this study, we report the approach employed to redesign and evaluate a general education course to enhance students' knowledge of tolerance and respect for diversity, drawing on Islamic values in the United Arab Emirates. In collecting and analyzing data for the study, we adopted a qualitative approach to explore students' nuanced and reflective understanding and application of the key concepts taught in the course. We retrieved and thematically analyzed 40 transcripts from sixty-nine students' reflective assignments. Our results indicate that students see the education process based on Islamic principles as a socialization means for shaping human life, caring for others, demonstrating compassion, and sharing knowledge, as essential to fostering tolerance. Drawing on our findings, we recommend carefully redesigning courses to strengthen students' knowledge of tolerance and respect for diversity using the authentic learning approach. This includes engaging them in activities to link class tasks to real-life contexts, providing them with safe learning spaces, and encouraging them to learn with peers from different backgrounds.

**Keywords:** tolerance; diversity; Islamic education; course redesign; undergraduate students; general education

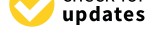



## 1. Introduction

With globalization bringing down societal borders, the need for respecting differences, practicing tolerance, and embracing diversity is essential for harmonious and mutually beneficial relationships in societies across the globe. More than before, people from diverse backgrounds, countries, cultures, and religions live, engage, learn, and work together. Today, more people are educated and expected to demonstrate civility, empathy, and respect for people within their communities irrespective of their varying demographic characteristics, including race, gender, and religious affiliation. Likewise, demonstrating the acts of tolerance and respect for diversity is now enshrined in the fabrics of societies and protected by laws—both locally and internationally.

The concept of tolerance is widely used in the context of diversity, particularly culture, gender, race, religion, and ethnicity. As a philosophical concept, tolerance is often described as the tendency to respect others irrespective of differences, being open-minded, and appreciative of cultural diversity. Such a conception has often been used in social psychological research and is included in the United Nations' Declaration of Principles on Tolerance (Verkuyten et al. 2019). Not surprisingly, Verkuyten et al. (2019, p. 8) describe tolerance as "acceptance despite disapproval, thereby keeping negative attitudes and beliefs from becoming negative actions." Yanti and Witro (2020) reported the existence of both false and accurate tolerance. In their view, false tolerance is characterized by pretense and

deception through which an individual may hide his identity in the presence of others with different orientations, backgrounds, and affiliations. On the other hand, accurate tolerance acknowledges, embraces, and respects differences and discourages discrimination against people for being different. Adding to this, Forst (2013) argues that tolerance requires the maintenance of specific standards and moral justifications for actions. Undertaking these endeavors with a section of society left out could be counter-productive. So, stakeholders in society need to work together in promoting tolerance through social control and education for citizenship at formal and informal levels.

As much as tolerance is critical for harmonious co-existence, discriminatory acts are still seen, experienced, and reported in societies. Recent studies indicate that intolerant and discriminatory actions are meted to innocent and law-abiding people on different grounds, including gender, race, religion, and social identity needs (Cvetkovska et al. 2020, 2021; Bagci et al. 2020; Yanti and Witro 2020). It is also reported that individuals within diverse societies tend to reflect on situations where they are marginalized without efforts put into the impacts of their privilege experience (Perez and Barber 2018). As the effects of these social problems affect societies at both micro (individual) and macro (community/state) levels (Tepperman and Curtis 2015), there is a need for stakeholders, governments, organizations, educational institutions, and individuals to intensify efforts in the fight against these issues.

In its bid to fortify its position as a diverse, heterogeneous, and accommodating nation that treasures diversity and tolerance, the United Arab Emirates (UAE) launched the Ministry of Tolerance in 2016, declared 2019 as the Year of Tolerance, and also inaugurated the Abrahamic Family House in 2019, among other initiatives. Like other countries, educational institutions in the UAE are expected to promote tolerance by instilling global citizenship values into curricula and modeling the practices in their policies and actions. Consequently, tolerance is emphasized in different subject areas, including Islamic education contents and values across different educational levels (Alhashmi et al. 2020). In the university contexts within the country, general education (GED) courses adopt interdisciplinary approaches to assist students in developing social and cultural values to be functional, ethical, and productive citizens (Aderibigbe 2020, 2021).

Alhashmi et al. (2020), however, report a shortage of research on curriculum contents promoting tolerance drawing on Islamic values in the UAE. Edwards (2018) also expressed the need for more studies focusing on strategies for reducing systemic oppression, including intolerance, discrimination, and a lack of respect for human dignity based on diverse backgrounds. We, therefore, responded to the calls by redesigning and exploring students' learning in the fundamentals of Islamic education GED course to strengthen students' knowledge of Islamic values for fostering tolerance, anti-discriminatory attitudes, and respect for diversity in a UAE semi-private university. We considered this move essential, as Islamic education promotes the values to build good societies and humans (Al-Attas [1980] 1999) in holistic and transformational manners (Alkouatli 2018). In redesigning the course, we also responded to the call for educators to rethink how GED courses are taught because of their potential to help students develop critical thinking skills (Rowe et al. 2015) and foundational knowledge to actively participate in the processes for making public-policies based on global thinking (Stubbs et al. 2018). Our findings complement the extant literature on strategies to redesign GED courses for meaningful learning and respect for diversity, leading to global citizenship among university students.

We present the related literature in the next section, followed by the methods section. We then report the findings, discussion, and conclusion in the last three parts.

## 2. Literature Review

### 2.1. Tolerance, Diversity, Justice, and Civic Engagement: Islamic Perspectives

Education as an agent of socialization plays a significant role in character building and the development of values of harmonious relationships and tolerance (Alhashmi et al. 2020), leading to good citizenship (Sampermans et al. 2021; Merry 2020). Perez and Barber (2018) also found formal education settings helpful in promoting intercultural competencies among

college students. In Islam, the concept of education revolves around three interrelated concepts—nurturing (*tarbiyah*), teaching and disseminating knowledge through instruction (*ta'lim*), and goodness or ethical behavior (*ta'dib*) (Al-Attas [1980] 1999; Yasin and Jani 2013). Explaining this further, Yasin and Jani (2013) contend that the educational process in Islam seeks to transform individuals into rational, spiritual, and social beings. In this sense, it aims to prepare individuals to be functional, ethical, and responsible members of society. So, developing knowledge in Islam is meant to stimulate moral, honest, and righteous practices that are grounded in spiritual consciousness. In this sense, it seeks to instill the spirit of global and effective citizenries into people irrespective of their social, cultural, economic, and geographical contexts.

An essential source of information in Islamic education is the Holy Qur'an. Islamically, it is believed that many concepts leading to respecting, tolerating, and accepting others are narrated and promoted in the holy book. For instance, verse 22 of Surah Al-Rum implies that humans should appreciate and embrace diversity of different types within their communities for peace to exist. The verse reads: "Additionally, one of His signs is the creation of the heavens and the earth, and the diversity of your languages and colours. Surely in this are signs for those of ⌐sound⌐ knowledge" (Translation by Mustafa Khattab). Surah Al-Baqarah verse 256 also clearly states that Muslims should respect religious tolerance and no one should judge others based on their religious affiliation. The verse reads: "Let there be no compulsion in religion, for the truth stands out clearly from falsehood. So whoever renounces false gods and believes in Allah has certainly grasped the firmest, unfailing hand-hold. Additionally, Allah is All-Hearing, All-Knowing" (Translation by Mustafa Khattab).

Muslims are expected to ensure that justice is maintained in any situation, irrespective of their relationship with those involved in the issues being addressed. Verse 135 of Surah Al-Nisa indicates thus: "O believers! Stand firm for justice as witnesses for Allah even if it is against yourselves, your parents, or close relatives. Be they rich or poor, Allah is best to ensure their interests. So do not let your desires cause you to deviate ⌐from justice⌐. If you distort the testimony or refuse to give it, then ⌐know that⌐ Allah is certainly All-Aware of what you do" (Translation by Mustafa Khattab). In Surah Al-Maidah verse 8, Muslims are admonished to be just and fair to others no matter how they feel about them. The verse reads: O believers! Stand firm for Allah and bear true testimony. Do not let the hatred of a people lead you to injustice. Be just! That is closer to righteousness. Additionally, be mindful of Allah. Surely Allah is All-Aware of what you do (Translation by Mustafa Khattab).

From the Qur'anic evidence shared (e.g., Surahs Al-Baqarah: 256; Al-Nisa: 135; Al-Maidah: 8) (The Qur'an n.d.), practicing accurate tolerance and fairness to all, regardless of differences, is described as the religious and moral duties for Muslims. However, Islam acknowledges tolerance not only for individuals but also for groups, through established codes of conduct. Surah Al-Hajj verses 67–69 state thus: "For every community We appointed a code of life to follow. So do not let them dispute with you ⌐O Prophet⌐ in this matter. Additionally, invite ⌐all⌐ to your Lord, for you are truly on the Right Guidance. However, if they argue with you, then say, "Allah knows best what you do. Allah will judge between you ⌐all⌐ on Judgment Day regarding your differences". (Translation by Mustafa Khattab). In this sense, Muslims must learn to respect laid-down regulations within their communities and tolerate others even if their opinions differ. Not surprisingly, Ibn Abbas reported in one hadith (sayings of Prophet Mohamed) that Prophet Mohammed, Peace Be Upon Him (PBUH), said, "Be tolerant and you will receive tolerance." (Aḥmad n.d.). In a reported Sunnah, the practice of Prophet Mohammed (PBUH), he was said to demonstrate tolerance and respect for differences when he was sitting with his companions and a funeral passed by, and in respect to the dead person, the Prophet (PBUH) stood. His companions were confused and could not help but notify him that it was a funeral of a person not of the Muslim faith. That is when the Prophet (PBUH), replied: "Was he not a soul?"

(Dawud n.d.). This noble act indicates that the Prophet came with Islam as a religion of liberty and respect for diversity by demonstrating true tolerance.

From the points discussed, it is not a gainsaying that differences are inherent characteristics of human societies and should be addressed through maturity and understanding (Yanti and Witro 2019). Today's social problems, such as poverty, intolerance, and racial injustices, are concerning and they can be resolved by embracing accurate tolerance, diversity, and people's differences (Yanti and Witro 2020). As enshrined in Islamic principles, all humans should enjoy a quality of life in society irrespective of their differences, including gender, religious affiliations, and race. Fostering the principles to promote tolerance and respect for diversity requires education, as discussed in the following sub-section.

*2.2. The Need for Tolerance in Education*

The increase in ethnic diversity in societies leading to increased cultural and religious differences comes with an increased duty to prepare students for diverse communities (Watson and Johnston 2006). Isac et al. (2018) also noted that increasing diversity is a global development, and its associated issues, such as intolerance, can be addressed through education. Indeed, education improves social life and ethical values to appreciate cultural differences and embrace tolerance, leading to empathetic and kind-hearted attitudes toward others, regardless of their differences (Sakalli et al. 2021). Therefore, developing active citizenship for participation and social cohesion is a core responsibility of educational institutions (Beemsterboer 2022), as societies need educated citizens with skills to make critical life decisions (Rowe et al. 2015).

As an essential component of education for character building, Lysenko et al. (2020) argued that tolerance is a moral philosophy that should be taught in schools to prepare students for a free, fair, and accommodating society. Not surprisingly, Lester and Roberts (2011) argued that knowing about world religions through educational courses enhances tolerance and respect for people's differences. Through education, students, as the future leaders, will develop the skills to tolerate and accept diversity when they are aware that societies may be characterized by their differences (Sakalli et al. 2021). Similarly, Grigoryeva and Grigoryeva (2020) argued that tolerance as an indispensable condition for harmonious and progressive co-existence in societies could only be promoted and sustained through education.

In order to address the different layers of the issues, it is argued that an education of tolerance should be variously developed at different societal and educational levels (Zakso et al. 2021). Sakalli et al. (2021) also contended that it is essential to structure educational programs so that students can develop the skills and knowledge to cope with others in the real-life world. In their study, which was focused on redesigning a general education course, Stubbs et al. (2018) argued that a course redesign offers the opportunity for curriculum content innovation. They also reported that their redesigned course assisted students in developing transferrable skills.

However, the literature indicates that students may not appreciate reorganized courses and the values of other programs apart from those floated in their departments. Head (2014) reported that students were unsure about the importance of some GED courses because of the perceived irrelevance of their contents to their core departmental programs. Rutledge and Lampley (2017) also found that changes in teaching strategy may not impact students' learning in some redesigned GED courses. Educators need to bear these reports in mind and be aware of the theoretical frameworks for understanding why people may engage in intolerant acts, discrimination, and other social problems. Understanding why people may demonstrate intolerant and discriminatory attitudes could assist educators when planning and educating students about tolerance.

*2.3. Theoretical Framework on Social Problems*

Intolerance and discrimination are examples of social problems as they are social conditions or behaviors that have negative impacts on many individuals within society, requiring stakeholders' concerted efforts to alleviate them (Tepperman and Curtis 2015). In

the literature, different theoretical perspectives on social problems are discussed, and we will explore the three commonly discussed perspectives as follows:

Functionalism holds that society requires social stability, which manifests through an appropriate socialization process and social integration. It also assumes that social problems weaken communities and could serve the interests of some individuals or institutions within society, which may put some individuals in vulnerable situations. (Durkheim [1897] 1952) argued that human desires might lead to chaos if they are not well-managed. In reducing social issues, social institutions, such as schools, religious centers, government establishments, and families, must ensure meaningful socialization, gradual reforms, and social integration for societies to function appropriately (Tepperman and Curtis 2015).

Conflict theory presumes that social problems result from societal power structures resulting in pervasive inequalities and conflicting interests (Wells 1979). The theory proposes an in-depth social change and societal structure to alleviate social inequality, reduce conflicting interests, and foster diversity (Tepperman and Curtis 2015). For this to be achieved, (Marx [1867] 1906) suggested that people negatively impacted by faulty societal structures may have to challenge the status quo and push back to reduce inequalities and other social issues. Doing this requires education to develop the skills to promote fairness and protect the rights of everyone in society.

Symbolic interactionism assumes that people learn in society through social interactions with others and understanding the interaction contexts (Blumer 1969). In doing this, the theory also presumes that individuals look out for symbols that help them interpret and understand social issues (Tepperman and Curtis 2015). So, people learn to perpetuate social problems, such as crime, discrimination, and racism, as they might have learned and developed certain attitudes through their interactions with others. Thus, societies can reduce social issues when peculiar root causes of problems are understood, and people engage with socially confirming individuals with respect for social controls.

As the theories indicate, people may perpetuate intolerant acts to serve their interests despite society requiring stability to function effectively from the functionalism lens. So, when people think about themselves at the expense of others, differences could be used as a basis for intolerance or discrimination, and this situation needs social change, as argued in the conflict theory. From symbolic interactionism, people learn through social interactions and symbols associated with their endeavors. So, efforts need to be intensified to ensure that the younger generations are exposed to activities and symbols that enhance their skills and embrace and promote diversity. Doing this effectively will require that educational programs are carefully planned and implemented using the teaching methodologies that offer students the opportunities to learn and apply the knowledge gained actively. This explains why we considered it a moral duty to redesign the fundamentals of Islamic education GED course to strengthen students' knowledge and skills for embracing diversity. In the next section, we explained how we redesigned a course and evaluated it to gain insights into students' reflections on the course.

## 3. Methodology

### 3.1. Purpose and Research Questions

In this study, we redesigned the fundamentals of Islamic education taught as a GED course to enhance students' knowledge of Islamic values and how they promote morality, tolerance, and respect for diversity. More importantly, our primary goal for conducting this study was to determine the extent to which the redesigned course assisted students in developing and strengthening their capacity to demonstrate tolerance and respect for diversity. Thus, we examined students' reflections on their learning and how the knowledge gained in the course could promote tolerance. Guiding the data collection process, we came up with the following research questions:

1. How do students conceptualize and interpret the values of Islamic education in the GED course?

2. What do students consider measures to treat everyone with respect and dignity in a diverse society?

3. How do students intend to use their knowledge of Islamic education to foster tolerance in society?

Figure 1 below depicts the framework and process for undertaking the research.

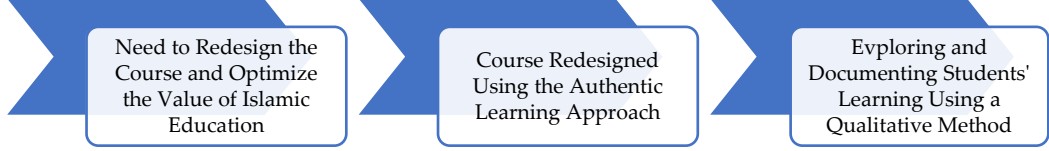

**Figure 1.** Research framework and process.

### 3.2. The Course and Context of Study

As earlier mentioned, the UAE intends to remain diverse and receptive to different ideas to sustain its growth trajectory and it expects institutions in the country to support this goal. We, therefore, recognized the need to enhance the process of instilling positive values in students as a critical moral enterprise for promoting tolerance and diversity in the country. We then redesigned the GED (fundamentals of Islamic education) course to allow students to engage thoroughly with learning resources, share ideas, and demonstrate an understanding of how best to apply the knowledge gained using the authentic learning approach. Authentic learning pedagogy is a teaching strategy that allows students to link class activities and real-life contexts in student-centered activities such as discussions, problem-based simulations, and scenario-based endeavors (de Lima 2021; Stanley 2021). It aligns with the notion that the knowledge transfer process in the 21st century is effective when it is dynamic and challenging, allowing students to adopt or discountenance ideas as necessary (Hussien et al. 2021). It is also grounded in constructivist principles, allowing learners to actively explore, engage, collaborate, and apply the knowledge gained, thereby taking ownership of the learning process (Aderibigbe 2020; Hsbollah and Hassan 2022; Hussien et al. 2021; Taber 2011).

We considered redesigning the course essential as it used to be taught for 14 weeks by focusing on core Islamic concepts and values with less emphasis on how students can apply the knowledge and skills gained to reduce social problems. In our view, teaching concepts and values without clear connections to specific real-life issues limits students' ability to apply their knowledge to reduce social problems that impact the diverse world we live in today. So, we modified the course description to include the application of values learned through the course to social issues.

Instead of exploring and teaching core Islamic concepts and values for fourteen weeks, we redesigned and divided the course into three modules. The first and second sections explored core concepts and theories, while the third focused on knowledge applications to social issues. We did this following the authentic learning principles to allow students to make adequate and explicit connections between class activities and real-life scenarios (Stanley 2021). Doing this also complements the need to emphasize critical thinking and inquiry practice in Islamic education-oriented classes (Hussien et al. 2021) and the two-way knowledge co-construction process among students (Aderibigbe 2011). Essentially, to achieve the course goals of enhancing students' knowledge of Islamic values and their place in reducing social problems such as intolerance and discrimination, we explored the following topics in three sections:

1. Theoretical perspectives and methods of teaching in Islamic education: concepts, objectives, and principles of Islamic education, information sources, and methods of teaching in Islamic education.

2. Ideological perspectives: Islamic culture, morality, pillars of Islam, and Iman.

3.  Application of values and knowledge gained through Islamic education in reducing social issues related to immigration and refugees, discrimination and racism, cultural variation, and diversity.

In teaching the course, the approaches used as components of the authentic learning process included interactive lectures, in-class and online discussions, storytelling, case studies, podcast analyses, and blended learning. On the other hand, assessments were performed using online discussions, short reflective essays, projects/case studies with presentations, and examinations.

### 3.3. Approach

We adopted the qualitative hermeneutic approach when conducting this study. Grounded in the interpretivist research paradigm that recognizes the subjective understanding and interpretation of realities, we employed the technique to explore, understand, and interpret students' subjective perceptions and texts (Creswell 2014). As argued, the priority of this strategy is to seek content understanding instead of quantification (Bryman 2016). Adopting this approach, therefore, assisted us in tapping into students' rich reflections and understanding of the redesigned course content to enhance their knowledge of Islamic values for promoting tolerance and diversity.

### 3.4. Ethical Considerations

We received institutional permission to conduct the study through the research context's college research committee. Then, we sought students' consent to participate in the study and reassured them of their right to withdraw without repercussions. So, we did not pressure students to participate in the study. Further, we only retrieved the texts of their participation in the assignments once their final grades were released, and we did not mention anyone's name in this report. By doing this, we ensured strict compliance with the ethical standards of scientific organizations.

### 3.5. Data Collection Procedure

All sixty-nine students enrolled in the two classes were considered as potential participants and were invited to participate in the study, with everyone indicating their willingness to participate. There were more females in the classes. Most of the students were between the ages 18 and 22, in the preliminary phase of their university programs, and were Arabs with a sound background in Islamic traditions.

When collecting data, we used the systematic sampling technique to select ten students at the count of five from each group to avoid the bias of choosing students who may be known to us. Doing this ensured that we had a manageable 40 transcripts instead of 276 from participants with similar demographic features and the relevant information to address our study's goals. More so, our intention was not to generalize the research findings but to seek rich data and valuable insights for understanding students' thoughts in the study's context (Aderibigbe et al. 2022).

We then retrieved the texts of their participation via two types of assignments in the course—three online discussions and one essay. The students were asked to engage with colleagues for the online discussions starting with their initial responses to questions posed and comments on at least two colleagues' posts. On the other hand, they were asked to complete an assignment on tolerance. Table 1 shows the given questions and the number of sampled texts.

As shown in the table, we collected 30 transcripts of the students' contributions to the online discussions and ten completed tolerance essays of between 1200 and 1500 words. Each student's response was between 150 and 200 words in each online forum task. These guidelines ensured that the students' contributions were well-thought-out and reflective of their understanding of the course.

**Table 1.** Tasks and sampled texts.

| S/N | Tasks | Sampled Qualitative Transcripts |
|---|---|---|
| 1 | Reflective Online Discussion 1: Drawing on your class discussions and readings, explain the concept of Islamic education. | 10 |
| 2 | Reflective Online Discussion 2: Based on the video clip watched in the class, discuss two Islamic values required for reducing discrimination and treating everyone with respect in a diverse society. | 10 |
| 3 | Reflective Online Discussion 3: If any, how would you use the knowledge and skills gained from this course in the future? | 10 |
| 4 | Tolerance Essay: Societies are increasingly getting diverse across the globe, and the need to live harmoniously cannot be over-emphasized. Against this backdrop, the UAE declares 2019 as the Year of Tolerance. Drawing on the sources of information in Islamic education discussed in your class, how would your group support the promotion of tolerance in the UAE society and beyond. | 10 |

*3.6. Data Analysis*

For the data analysis, we used an inductive thematic analysis approach by uncovering common and repeated patterns of ideas and meanings in the textual data (Saldaña 2013). In doing this, we used the NVivo qualitative analysis software to organize the data into codes. We then reviewed the codes and generated the themes drawing on substantial elements of the raw texts. After grouping the codes to create themes, we checked the themes and codes thoroughly to determine the extent to which the patterns represent students' reflections and thoughts. As an interdisciplinary team, the iterative process through which we read and validate the patterns emerging from the findings lends credibility and dependability to the study (Lincoln and Guba 1985).

**4. Results**

In this section, we present our results drawing on the students' reflections on the re-designed course and how the knowledge from the course promotes tolerance within diverse societies. In describing the results, we used the research questions as a framework and presented the themes that address the questions, along with related supporting vignettes.

*4.1. How Do Students Conceptualize and Interpret the Values of Islamic Education in the GED Course?*

Our first research question was to determine the extent to which students see values in the fundamentals of Islamic education as a GED course. In response to the question, three themes emerged from the data analyzed, and they are highlighted as follows:

4.1.1. A Learning Process Grounded in the Teachings of the Quran and Sunnah

Reflecting on the course content, students indicated that Islamic education differs from the conventional learning process as it predominantly draws on the teachings of the Quran and Sunnah. For example, one student explains the following:

*The content used for Islamic education is derived from the Quran, the Sunnah and authentic Ahadith* (Group 1, Discussion 1, Student 2).

Sharing the same sentiment, another student shares the view below:

*Islamic education is simply different from other types of educational theory and practice largely because of the all-inclusive influence of the holy Quran. The holy Quran serves as*

*a comprehensive plan for both the individual and society and as the primary source of knowledge* (Group 2, Discussion 1, Student 1).

Explaining this further, a student contends the following:

*Islamic education comprises a set of concepts and tenets pertaining to human nature, creed, intellect, and attitude, along with spiritual and physical values, all entwined in unified perceptual framework and relying, entirely, in its fundamentals and morals on the Holy Quran and the Prophet's Sunnah* (Group 1, Discussion 1, Student 5).

As much as students agree that the principles and philosophies shared in Islamic education are influenced by the Quran and Sunnah, they also feel it touches other elements and fields within society. This sentiment is enunciated as follows:

*I think that Islamic education contains three different fields of learning, the first is understanding the Quran and Sunnah then apply these teachings either as a belief system or as a way to behave with other people be it Muslims or non-Muslims* (Group 1, Discussion 1, Student 3).

### 4.1.2. An Educational and Socialization Process for Shaping Human Life

For this theme, what brings the students' statements together is the reference to Islamic education as being relevant to someone's *life*, as opposed to some specific situations that a person may find themselves in. For example, in the following passage, a student believes that Islamic law should be taught so as to shape one's lifestyle:

*I believe it aims to teach the younger generation about the Islamic principles in an effective method for them to turn it into a lifestyle and to encourage Islamic education as it is crucial for the people to fully understand and implement the Islamic law to live peacefully* (Group 1, Discussion 1, Student 2).

This sentiment is reinforced by other students in the following passages:

*Islamic education is a lifestyle that every Muslim should follow* (Group 2, Discussion 1, Student 2).

*Islamic education includes many things and has many dimensional that links in the Individual's life* (Group 1, Discussion 1, Student 1).

### 4.1.3. A Cultural Learning Process to Imbibe and Promote Islamic Values

This theme concerns descriptions of Islamic education as being related to a cultural learning process through which one can develop, internalize, and promote Islamic values. In forming this theme, one student describes how Islamic education is constituted around learning Islamic values:

*Islamic education shapes the person in a manner which enables them to connect their actions, decisions and attitudes towards life to Islamic values* (Group 1, Discussion 1, Student 2).

The expression of the theme in terms of Islam's values being what concerns Islamic education is also demonstrable in one student's reference to properly treating others as being key to Islamic education:

*Islamic education is not only about praying 5 times a day or just fasting in the month of Ramadan, it's also about one's "Akhlaq", as the Prophet Muhammad (PBUH) said, the best among you are those who have the best manners and character* (Group 1, Discussion 1, Student 3).

### 4.2. What Do Students Consider the Measures to Treat Everyone with Respect and Dignity in a Diverse Society?

In addressing this question, two major themes emerged from the data analyzed, encapsulating students' thoughts based on their understanding of the course. The themes are enunciated as follows:

### 4.2.1. Demonstrating Respect and Non-Discriminatory Attitudes

The students who expressed this theme demonstrated, in varying degrees of directness, an understanding of Islamic values as promoting relations between groups that are non-discriminatory.

*Treating people and one another with respect and dignity is not an option to choose from, it is basically concepts that we should live within regardless sex, age, race or nationality, etc. Before defining those words, we should know their cores as they are coming from fairness, respect, equality, dignity and autonomy (FREDA). People should teach their children to grow up within this thinking whether from an Islamic background or even other religions these are just basic human rights and a way to treat one another* (Group 1, Discussion 2, Student 3).

*Respect includes loving for our brothers what we live for ourselves. Respect involves treating others the way we wanted to be treated with love, compassion and mercy* (Discussion 2, Group 1, Student 5).

Adding to respect for others, students indicated the need to demonstrate non-discriminatory attitudes toward others:

*Our religion teaches us to respect everyone whether a Muslim or a non-Muslim, rich or poor, good human or no maybe seeing your good behavior may result into a positive change* (Group 1, Discussion 2, Student 5).

*Islam has urged Muslims to contribute in stopping any forbidden act by either actions, words or heart. So whenever you see and act of discrimination or bullying for being part of a certain group, conduct the highest level of advice which is taking an action, if not advice by tongue or pray for them* (Discussion 2, Group 2, Student 4).

### 4.2.2. Being Forgiving, Caring, and Empathetic toward Others

As opposed to just describing the importance of respect or how anti-discriminatory acts are important social tenets, the students indicate the essential place of caring and empathizing with others in Islamic values. The words of a student are as follows:

*As Muslims, our religion teaches us to forgive and forget whenever someone mistreats you for being different, this makes the other person feel guilty that you faced their bad treatment with high manners, then they wonder how good you were raised and on what rules that they become interested to learn and avoid discriminating others* (Group 2, Discussion 2, Student 4).

Another student highlights the need to support people in society, irrespective of their religious orientation and ethnic background. Reflecting on a video clip watched in the classroom, the student wrote the following:

*One story that was shared in the video was about a woman named Jodie who had gone homeless and was struggling to find food for herself and her husband but was gratefully saved by the 'Bearded Broz' when they continuously offered them food until they finally landed back on their feet. The result of their kindness allowed Jodie to volunteer regularly and help deliver food to other homeless people. The commitment and consistency in this video really demanded the attention of accepting others and not allowing your illusion of a perceived religion to cloud your judgment* (Group 2, Discussion 2, Student 2).

Acknowledging the need to be supportive and empathetic to others, a student contends that learning and raising awareness about social issues are responsibilities of everyone:

*After watching the video [on domestic worker abuses] we need to know more about domestic workers. We also need to raise the awareness of the rights of domestic workers, they must have a law that protects their rights in the worldwide and I'm sure that UAE law is giving each person who lives on its land their rights* (Group 1, Discussion 2, Student 2).

Reinforcing the argument, a student wrote the following:

*Treating people and one another with empathy and dignity is not an option to choose from, it is basically concepts that we should live within regardless sex, age, race or nationality, etc.* (Group 1, Discussion 2, Student 3).

4.2.3. Setting Rules and Guidelines

The students also indicate the need for setting rules as a measure to reduce intolerance and cultural barriers within diverse societies. This is enunciated as follows:

*Set rules that protect individual from suffering of any type of problems related to diversity. In the workplace, diverse teams usually make better decisions; they are more creative and attentive to your primary goals and customers' needs... Policies, procedures, safety rules and other important information should be designed to overcome language and cultural barriers by translating materials and using pictures and symbols whenever applicable* (Essay transcript 4).

Supporting this sentiment, another student referenced a UAE legislation focusing on the promotion of diversity and true tolerance. The students also made a connection between the gesture and the provision of the Holy Qur'an in relation to the promotion of diversity:

*UAE President His Highness Shaikh Khalifa Bin Zayed Al Nahyan gave Federal Decree No 2 of 2015 on Combating Discrimination and Hatred. The law denounces oppression of people or gatherings dependent on religion, regulation, conviction, statement of faith, rank, race, shading or ethnic source. This law upholds the society's structure and cultural diversity that is stated clearly in the Holy Quran: "To every People have We appointed rites and ceremonies which they must follow, let them not then dispute with you on the matter, but do invite (them) to your Lord: for you are assuredly on the Right Way. If they do wrangle with you, say, 'God knows best what it is you are doing. "God will judge between you on the Day of Judgment concerning the matters in which you differ.'" (Quran, Al-Hajj: 76–69)* (Essay transcript 5).

*4.3. How Do Students Intend to Use Their Knowledge of Islamic Education to Foster Tolerance in Society?*

Lastly, we wanted to know if students see the knowledge gained from the course as applicable to promoting good citizenship and tolerance with positive impacts within diverse societies. For this question, three themes emerged from the data analyzed, and they are presented as follows:

4.3.1. To Be Kind and Helpful to Others

The students expressed that they would behave in such a way that they will be kind, helpful, or good to others. For example, a student wrote about being encouraged to be kind to everyone, in relation to the Islamic education course:

*This course talked about general things which is most needed to get through life. Real life situation and how we should look at them from an Islamic prospective. We talked about migration and refugees. It gave me knowledge that there are people who have left their homes not with their happiness. Therefore, it encourages me to be kind and polite with everyone since we don't know what they were going through* (Group 1, Discussion 3, Student 5).

This content theme took another form: it was also expressed in terms of one student describing being reminded to be kind and put smiles on others' faces. Specifically writing as follows:

*Drawing a smile on everyone's face, be kind and gentle to all people without segregation* (Group 2, Discussion 3, Student 3).

Another student supports this sentiment, acknowledging the need for such acts in today's globalized world, as follows:

*In this globalization, it has become more essential and significant to maintain tolerance and harmony, and to promote shared unconditional love and affection, when people with different backgrounds, cultures and faiths coexist and in a world that is becoming more multicultural and diverse than ever* (Essay transcript 3).

### 4.3.2. Sharing Information and Knowledge with Others

The most widespread response to the question was that of students describing how they would share knowledge with others. This is enunciated in the following passages:

*The way I would use the knowledge I gained in this course is by sharing it with the people I know* (Group 1, Discussion 3, Student 4).

*It motivates me to spread Islam from my behalf without a professional location. It can be done anywhere if you have the correct information* (Group 1, Discussion 3, Student 5).

In reinforcing this sentiment, students believe it is essential to meet and engage with people from different backgrounds to learn and share knowledge:

*As a Muslim you have to seek for knowledge through meeting and interacting with people from different background and religious beliefs. Knowledge gained can be shared by you with people you like friends or family* (Group 2, Discussion 3, Student 3).

Additionally, students acknowledge the need to teach the value of tolerance and respect for diversity to children as a way of sharing information to instill positive impressions about tolerance to them, as that will positively impact their adult lives:

*His highness sheikh Mohammed bin Rashid AL Maktoum said "There is no future in this region without intellectual restructuring that consolidates the values of tolerance pluralism and acceptance of others intellectually, culturally and in terms of sect and religion* (Essay transcript 5).

*If we need a generation that tolerates their fellows, we need to start teaching kids at a young age how to tolerate others. The more they grow up and their senses and understanding develops, they start realizing how to practice tolerance in every aspect of their life, whether with their spouses, their friends, their siblings and even strangers* (Essay transcript 2).

### 4.3.3. Be Tolerant and Accommodating

Expressing this view, a student described how they themselves will behave in a non-discriminatory or tolerant way as a result of the course:

*This course taught me do not judge anyone's opinion respect everyone's opinion because everyone has different opinion and idea, and we can't judge them and force them to accept our idea* (Group 2, Discussion 3, Student 4).

Another student explained that they have been reminded by the course to be tolerant or non-discriminatory:

*This course is a reminder for me as a Muslim on how a muslin should act toward others, to respect people with different beliefs, to not accept them nor believe in them; instead get to know others culture through respecting their ideas and thoughts* (Group 2, Discussion 3, Student 3).

Another student said that they will share knowledge pertaining to non-discrimination and tolerance, as a result of the course:

*The way I would use the knowledge I gained in this course is by sharing it with the people I know. With the knowledge gained in this course I would encourage people to be better persons and give esteem to everyone without judgement because of how they might look or practice. Be fair and tolerant to others* (Group 1, Discussion 3, Student 4).

## 5. Discussion

In this study, students appreciate the value of the redesigned GED course and see it as essential for building and strengthening their knowledge of tolerance and respect for diversity. This is at variance with some studies indicating that students do not treasure the GED courses as their curricular contents are irrelevant to their professional growth (Aderibigbe 2021; Head 2014) and changes in pedagogical approach (Rutledge and Lampley 2017). As the data revealed, the students' reflections indicate that the course assisted them in assimilating values for becoming responsible humans with positive attitudes toward others, regardless of their differences. Perhaps this explains why functionalism holds that stakeholders need to provide appropriate socialization and social integration for societal stability (Durkheim [1897] 1952). Alhashmi et al. (2020) also indicated that education drawing on Islamic values assists students in developing good characters and attitudes, including an appreciation for differences, tolerance, and peaceful co-existence. Developing such attitudes leads to good citizenship (Sampermans et al. 2021; Merry 2020). In the same vein, Perez and Barber (2018) reported that students were able to develop intercultural knowledge through formal education contexts. Not surprisingly, the students conclude that building normative behaviors and knowledge in the course is grounded in the Islamic teachings that are exemplified in the Holy Qur'an and Sunnah. In Yasin and Jani (2013), the educational process in Islam is also said to assist individuals in becoming rational, ethical, and spiritual. Therefore, this finding reinforces the need for tolerance and respect for diversity to be taught in all appropriate curriculum contents, drawing on religious and cultural values in schools as an essential agent of socialization. Considering this endeavor as a moral enterprise ensures an effective process to build good communities and individuals (Al-Attas [1980] 1999).

As the data also showed, students indicate that tolerance and diversity can be fostered by accepting and discharging civic responsibilities in society, including setting acceptable standards (rules) and acknowledging cultural boundaries. For them, it is essential to respect others and act in non-discriminatory ways. In so doing, individuals need to demonstrate an understanding and appreciation of differences as inherent societal characteristics and relate to each other with maturity (Yanti and Witro 2019), leading to a harmonious co-existence. Doing this will also ensure that using discriminatory acts to promote some interest at the detriment of others should be jettisoned (Durkheim [1897] 1952), and that inequalities of any kind are discouraged because they could trigger social problems (Marx [1867] 1906). However, Forst (2013) contends that accepting others does not stop one from upholding specific standards. So, it is essential to practice authentic tolerance where individuals show appreciation to others but showcase their culture and beliefs without pretense (Yanti and Witro 2020). Indeed, authentic and genuine tolerance should be desired and encouraged as a way to enhance peaceful, respectful, and harmonious co-existence in diverse human contexts.

Similarly, students feel forgiving, caring, and empathetic toward others, which are essential civic responsibilities that are capable of promoting harmony and respect for the dignity of people, regardless of their backgrounds and differences. Imbibing these values is undoubtedly a virtue for demonstrating good character, as exemplified in the Prophet Mohammed's (PBUH) noble act of mourning and sympathizing with non-Muslims during his time. The Prophet's utterance that was he, not a soul, when his companions were confused that he showed respect to a dead non-Muslim person indicates the need for everyone to be appreciated and loved irrespective of their backgrounds, races, or religious affiliations. So, people learn through interaction and symbols, including the words and actions of their mentors (Blumer 1969). Alhashmi et al. (2020) also contend that knowledge of Islamic values promotes cooperation and respectful civic engagement through critical thinking and reflections. GED courses are also said to employ a multidisciplinary approach to instill positive values into students, leading to functional and productive citizens (Aderibigbe 2020, 2021). However, to promote critical thinking and reflections for strengthening tolerance and respect for diversity, it may be essential to adopt the inspirational

pedagogies that draw on practical actions taken by inspirational figures that are adored by students within specific and international cultural contexts.

It Is not a surprise that the students indicate that they have roles to play in promoting tolerance and respect for diversity, drawing on the knowledge gained from the course. For instance, they acknowledge the need to be kind and helpful to others, which reinforces the previous finding concerning empathy, and can be likened to *ta'dib*—goodness and ethical behavior (Yasin and Jani 2013). The students also feel that they will be more tolerant and accommodating to differences as they hope to demonstrate these virtues and share the information to help others develop the knowledge and skills to treasure diversity. As Yasin and Jani (2013) pointed out, these values are critical elements of education based on Islamic principles—nurturing (*tarbiyah*) and seeking and disseminating knowledge (*ta'lim*). By doing this, students will complement the government and educational institutions' efforts in reducing the social problems (Tepperman and Curtis 2015; Alhashmi et al. 2020) that are associated with intolerance and discriminatory acts, which could have micro (individual level) and macro (community/global level) impacts within societies.

## 6. Conclusions

In this study, we analyzed students' reflections on the fundamentals of Islamic education course, which was redesigned to enhance their knowledge for fostering tolerance and respect for diversity. Based on the outcomes of our analysis, it is clear that redesigning courses to assist students in gaining practical and meaningful learning experiences is an indispensable and noble act for educators in the diverse world that we live in today. Teaching students the values to promote tolerance and respect for diversity as essential civic responsibilities should be a moral obligation and priority of educational institutions. Essentially, we contend that raising awareness about tolerance and respect for diversity that draws on cultural values as the embedded curriculum contents should not be jettisoned but embraced as a necessity to promote harmony in diverse societies.

To foster the educational process for enhancing students' knowledge about Islamic values, including tolerance and respect for diversity in other contexts, we recommend redesigning courses using the authentic learning approach to provide meaningful learning opportunities for them to demonstrate the knowledge gained and link class activities to real-life situations. These include adopting critical and constructivist pedagogies such as social analyses, online discussions, case analyses, podcast reviews and analyses, problem-based analyses, applied projects, and presentations. Additionally, educators should provide safe and secure learning spaces for students to respectfully engage in possibly sensitive but honest conversations about tolerance and the associated social problems. Further, educators should encourage students to learn and develop friendships with peers from diverse backgrounds without flouting any cultural, institutional, or state boundaries and regulations. Doing these things will enhance their critical and problem-solving skills to reflect on how intolerant attitudes, negative tolerance, and resentment toward diversity can negatively impact communities.

**Author Contributions:** Conceptualization, S.A.A., M.I., K.A., H.A., W.B.H. and A.A.C.; methodology, S.A.A., M.I., K.A., H.A., W.B.H. and A.A.C.; formal analysis, S.A.A. and A.A.C.; investigation, S.A.A. and M.I.; resources, S.A.A., M.I., K.A., H.A., W.B.H. and A.A.C.; data curation, S.A.A.; writing—original draft, S.A.A., H.A., W.B.H. and A.A.C.; writing—review and editing, S.A.A., M.I. and K.A. All authors have read and agreed to the published version of the manuscript.

**Funding:** This research received no external funding.

**Institutional Review Board Statement:** This study was conducted in accordance with the Declaration of Helsinki, and was approved by the Scientific Research Committee, College of Arts, Humanities and Social Sciences, University of Sharjah.

**Informed Consent Statement:** Informed consent was obtained from all subjects involved in the study.

**Data Availability Statement:** Not applicable.

**Acknowledgments:** We thank the students who participated in this study, and are indebted to the University of Sharjah for providing an enabling environment for us to undertake the study.

**Conflicts of Interest:** The authors declare no conflict of interest.

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
