# Peer review of "Fostering Tolerance and Respect for Diversity through the Fundamentals of Islamic Education"

_religions, doi:10.3390/rel14020212_

Round 1

Reviewer 1 Report

Dear Author/s,

Thank you for sharing your interesting work. The article undersells your work, unfortunately. The introduction, literature review and theoretical framework are weak and do not justify your research questions. Your methodological choices are also insufficiently justified. Some critical distinctions are not adequately dealt with, or not dealt with at all. I believe that this article needs a lot of reworking before it accurately reflects what you did and why you did it. I encourage you to engage critically with other literature to support and enrich your work.

Thank you and all the best.

Author Response

We sincerely thank you for reviewing our work and providing valuable feedback. We have revised our paper's introduction, literature review, and methodology sections based on your advice. We also corrected and revised some parts of the findings, discussion, and conclusion sections as advised. We hope this revised version is okay, but please let us know if we need further modifications to enhance the paper.

Response to annotated manuscript are as follows:

Abstract

·       … as a socialization means corrected.

·       … invoked… revised.

·       … argued that and the transition following the sentence revised.

·       … melted changed to meted.

·       In its bid… paragraphed.

·       Correspondingly changed to consequently.

·       University electives are described as General Education (GNED) courses in our research context.

·       Highlighted XXX, 2021 is to hide authors’ identities for peer review.

·       The scholars, however… changed to AlHashmi et al (2020).

·       …we collect… deleted and replaced with we responded…

Literature review

2.1. sub-section

·       Taleem’s description revised.

·       All the citations from the Quran revised and supported with translation information.

2.2 sub-section

·       We added the sub-section as advised.

Sub-section 2.3

·       We revised the sub-section and added more references where necessary.

·       We explained how we came to the conclusion on the sub-section.

·       Course redesign transferred to methodology section.

·       Authentic learning transferred and discussed in methodology section.

·       Research questions are represented in methodology section with more clarity on how the literature informed the questions.

Methodology

·       We reorganized and revised the entire section.

·       We clarified how the course was taught in the past.

·       We added more information about the aim and goals of the course.

·       We explained what pedagogical approach means with examples.

·       We discussed the choice of strategic sampling and how we used it.

·       We added a sub-section on ethical considerations, as advised.

·       We provided more information on how we analyzed and interpreted the data. We also clarified how the quality of students’ submissions were determined in the preceding data collection procedure sub-section.

Results

·       We presented students’ reflection and perspectives based on what we had taught them or what they have learned in the course.

·       We corrected the inconsistent quotations.

·       We revised the connection mentioned in 4.3, as advised.

Discussion

·       We added the references highlighted to the literature review section (sub-section 2.2).

Conclusion

·       We revised the highlighted sentence with the word enlightening.

Reviewer 2 Report

This is an interesting paper that deserves the full attention of the Islamic world. Being Muslim is more than a basic religious attitude, it also presupposes a vision and an action perspective on the good life. This is well reflected in this paper. Tolerance is taught as a religious and moral duty in Islamic Education. Here, not only the Qur'an, but also the Hadith, the Sunnah, and the "consensus" is of inspiring importance.

The theoretical framework on social problems and the role of education is well presented, there is talk of an "authentic learning pedagogy," which "allows students to link class activities and real-life contexts" in GNED (Fundamentals of Islamic Education) course. There is talk of not just non-discriminatory behavior, but also pro-social behavior. However, the question remains what concrete learning measures are taken to put this hermeneutic and didactic concept into practice. With some concrete examples of action models and a description of the learning context in which they were offered, the paper becomes even more powerful. A second question is that of inter-religious learning: does this concept also offer the opportunity to reflect and act on themes such as compassion and pro-social behavior in a multi-religious context?

Author Response

We thank you sincerely for your kind remarks and the time taken to review our paper. In response to your valuable feedback, we provided examples of class activities to promote students’ knowledge and application of pro-social attitudes in the methods and the findings section. We used the demonstration of tolerance and respect for diversity to show students’ demonstration of pro-social behaviors. We also used diverse societies and different backgrounds across the paper to depict the diversity of different types, including religious diversity. We have revised all the sections of the paper considering all the feedback received. We hope this revised version is okay, but please let us know if we need further modifications to enhance the paper.

Reviewer 3 Report

The aim of the research is noble, but the paper is weakly structured on the basis of scientific guidelines. The context / reason of the research is not written. The theoretical framework is not functional. The dates should be described in more detail. The analysis method is not well described. How the conclusions were reached is unclear. In the discussion part I do not read a discussion but a repetition of the findings.

Author Response

Thank you so much for reviewing our paper and providing valuable feedback. Based on your advice, we have revised all the sections of the paper. For instance, we added a sub-section to the literature review section and reworked the other sub-sections. We also revised the discussion and conclusion sections. We hope this revised version is okay, but please let us know if we need further modifications to enhance the paper.

Response to annotated manuscript are as follows:

Abstract

  • We clarified the participants’ number and study context, as advised.

Introduction

  • We indicated that the study was conducted in the UAE.

Literature Review

Section 2.1

  • We did not add the page number because the idea was paraphrased and not quoted directly.
  • We clarified that Islamically, many concepts leading to tolerance are explained and promoted in the Holy Quran.
  • We clarified that Hadith and Sunnah are different.
  • We revised the sub-section to show that not all the readers of the work are likely to be Muslims.
  • We also presented Quranic verses for promoting tolerance. Islamically, it is believed that the content of the Holy Quran was released in batches and there are some verses sent during the war time in the past.

Section 2.2

  • We revised the section and indicated how it contributes to the paper. We also used the sub-section to discuss our findings in the discussion section.
  • We explained what the authentic learning means in the methodology section.

Methodology

  • We revised and reorganized the entire section.
  • We used Iman instead of Eeman based on Khattab’s translation.
  • We revised the data collection process.

Results

  • We used about 6 or 8 video clips for case studies in the course. However, we selected one of them in the tasks analyzed. Using the clip allowed students to analyze a case and relate their studies to the scenario.
  • We indicated that FREDA is an acronym and presented it after the words/meaning.
  • We corrected the inconsistent quotes of students’ texts in bold.
  • We addressed the repetition of texts and represented the ideas without duplications.

Discussion and Conclusion

  • We revised discussion to show what we found along with the extant literature and possible implications of the results.
  • We revised the conclusion section as well.

Reviewer 4 Report

Comments to the Authors

Overall, this is a creatively-conceived and -designed study that will contribute to various bodies of literature, including Islamic Education Studies. It is generally well written and clear. It comes at an interesting time, when all the world’s attention is focused on the Gulf region. This also comes with some sensitivities that the authors need to unpack in the text, as I noted below. The paper is important and timely. Yet, I have some substantial comments below, per section, which I offer in the hopes of making the paper stronger.

But, first, two overarching points: 

1. Given the paper’s central concern with the role of Islamic education in fostering tolerance, and the related potential contribution to the emerging field of Islamic Education Studies, the authors are missing many key scholars within that field. The authors would do well to go back through that literature and look up classical names like Naguib Al Attas, Timothy Winter, Sayed Hussein Nasr, along with contemporary names like Abdullah Sahin, Farah Ahmed, Claire Alkouatli, Nadeem Memon. Even if they do not choose to include these scholars, they need to engage more deeply with the literature on Islamic Education Studies and maybe even describe the disciplinary location of this paper as being within that field (unless there is another more pressing field that they should be located within; either way, they should specify). They also may want to draw upon the Islamic Education Studies scholars mentioned above  in establishing a stronger Islamic conceptual frame, along with quotes from the Qur’an, as well as scholars from Islamic studies.

2. A perspectival shift is needed, which is timely and difficult to offer, but I think that these authors have the ability and the data to do it: Rather than attempting to match Islamic values to the ideals and objectives of social cohesion defined by these non-Muslim scholars, the authors can note overlaps but ultimately an Islamic paradigm of sociocultural character values would extend further than the one set out by the UN, which they call “global citizenship values” and have drawn some critique for this.

Now I will detail the areas that need attention section by section:

Abstract: the authors need to mention the UAE context from the outset; also mention number of participants and data sources, and analysis etc. 

Introduction

The authors set up a nice introduction discussing a diversifying world and what nations are doing about it, zeroing in on the UAE, and then taking a next logical step to present why education is important for teaching tolerance. They then establish a beautiful little gap in the literature, as added justification (because a gap alone is never sufficient as a stand-alone rationale): scholars have reported “a shortage of research on curriculum contents promoting 73 tolerance drawing on Islamic values in the UAE.” All of this is great. 

But then comes a disconnect prior to introducing the course: you need to tell us why did you chose this course? Why does it need to be redesigned. 

“In this study, we report how a GNED (Fundamentals of Islamic Education) course 78 was redesigned to enhance students’ authentic knowledge of religious and cultural values 79 to promote tolerance within diverse societies.” 

The authors should describe why they focused upon this course to redesign? Why is it the vehicle of tolerance education in the UAE? 

In addition, there is one more important step that could be taken to really nail the rationale: to realize the potential of Islamic education for character development and unity across diversity, which is currently unrealized. In other words, the second part of the rationale would be a critique of contemporary Islamic education in that the way it is being taught does not do justice to its potential for human beauty of character. The platitudes offered in the beginning of the paper—like “globalization bringing down societal borders”—is only a first step towards a rationale; there is one more step to take. And that second step would be a real contribution to the literature. 

The authors should try to finesse this four-part rationale: 1. Diverse world requiring tolerance 2. Nations focusing on tolerance including UAE 3. Lack of literature and courses in UAE on tolerance and 4. Immense but under-utilized potential in Islam/Islamic education for providing a comprehensive scope of tolerance education.

If the authors can do this, it will help clarify the boxes in Figure 1, where each box should be explained. As of now, box 1 does not have strong enough justification for why this course needs to be redesigned.

2. Theoretical Frame and Literature Review

The literature review and the theoretical frame are mixed up! The concepts on tolerance and diversity in the Qur’an constitute the real conceptual frame that the authors should then employ to both re-design the course and evaluate the outcomes. This below seems to indicate the real conceptual frame:

“An essential source of information in Islamic Education is the Holy Qur’an. All the concepts leading to respecting, tolerating, and accepting others are narrated and explained in the holy book. For instance, diversity is acknowledged and encouraged in Qur’an.”

As it is now, the authors set a theoretical frame of western, non-Muslim contexts rooted in a secular paradigm (and they actually never justify why this framework is needed) in order to redesign, via authentic learning pedagogy, a course to enhance Islamic values: it doesn’t make sense; it is like the ground to stand on is missing. The authors need to start with an Islamic conceptual frame to ground the re-design and evaluation. Then they can back it up with all the Marx perspectives and UN global values that they want. But they have to have a solid and relevant frame first.

After they establish the Islamic conceptual frame, the literature review can then include a section on the western tolerance scholars, but an even more important section would be what previous Muslim scholars have said about tolerance in the context of Islamic education. This is currently missing from the Lit Review.

3. Methodology 

The methodology is relatively sound but there are a couple of questions, including about analysis: I am wondering why the authors needed NVivo on such a small sample and I’m also wondering how they triangulated the data and/or engaged in collaborative analysis between themselves. How many researchers were conducting the analysis, who did what, and how did they work to deepen and extend each other’s’ work? 

4. Results

This entire Results section needs a higher-level analysis—a total re-analysis—because it requires deeper thought and clearer organization. At this point, it seems the authors are simply listing the ‘interview’ questions and the best of the students’ responses. Two related general qualitative principles are that a) the research questions should not be the same as the interview questions; in many cases they cannot be, because the research questions are overarching, very nuanced, and articulated in language aimed at academic colleagues, whereas the interview questions are clearly articulated in lay language, more digestible, and aimed to together answer the research question; and b) the themes cannot be the research questions. The research questions could be used to organize the themes, and this should be clearly seen and stated. But the titles in the Results section should be the theme titles. Then you can in a summary paragraph answer all of them together. 

So, in other words, although it is always very satisfying when a researcher answers the research questions at the end of a paper, the research questions should not constitute the headings of the results: instead, the headings should be the themes the authors’ discerned during analysis. Otherwise, it’s not really analysis: it’s simply reporting answers to questions. Even more reductionist, it seems that for each research question asked of participants they were given a specific task: “For this question, students were invited to watch a video clip” […]. This should be rewritten.

A third general principle is that every data excerpt should be layered with analysis: in other words, it is optimal to always have some analytic text before but especially after a data excerpt, to explain, contextualize, unpack it. This is needed throughout, especially in the sensitive or controversial sections! For key examples that need further analysis and unpacking are, the excerpt of “UAE President His Highness Shaikh Khalifa Bin Zayed Al Nahyan” and section 4.2.2: “We also need to raise the awareness of the rights of domestic workers, they 419 must have a law that protects their rights in the worldwide and I'm sure that UAE law 420 is giving each person who lives on its land their rights”

The employment of the same names for different themes and sections happens at least twice in the Results and is very confusing and indicates that more analysis is needed. Technically, two themes that are the same suggest that they are actually one theme. The below needs to be fixed:

—Section 4.1.2 (An educational and socialization process for shaping human life) is the same as 4.2.1. (Socialization process for shaping human life) 373 

—Section 4.1.1. A learning process grounded in the teachings of the Quran and Sunnah 309  is the same as 4.2.3. A learning process grounded in the teachings of the Quran and Sunnah 427 

Discussion

Excellent how the authors circle back to the framing literature. They could even make the point more clearly: “drawing on religious and cultural values in schools as an essential agent of socialization” to state that exploring Islamic education and values in action is required for character development in this context. And, as noted above, involve a wider variety of scholarly voices from the field of Islamic Education Studies. 

Small Additional Comments

—A Jewish person, or a person of the Jewish faith or a person not of the Muslim faith, is a better term than ‘a Jew.’

—The ethical notes on page 7 are good, but you should tell us how you secured ethics approval overall: from what institution? And also from the school itself?

—Watch the random and inconsistent use of capital letters: Being forgiving, Caring, and Empathetic towards Others 

—Please don’t use this acronym after the name of the Prophet Muhammad, peace and blessings be upon him! The letters are so jarring: PBUH. Instead, could you add the Arabic symbol? This is what I always do and mainstream publishers accept it in the name of cultural relevance, and its more beautiful!

Author Response

We sincerely thank you for reviewing and offering valuable feedback on our paper. We revised the paper based on the comments of the previous three reviewers and got to know of the fourth reviewer when we were about to submit the revised version. We have made some changes based on your advice and can also confirm that some changes have been made based on the feedback earlier received to enhance the paper. We hope the revised version is okay, but please let us know if we should revise it further.

Round 2

Reviewer 1 Report

Dear Authors,

Thank you for making some revisions to your article. However, on reading the revised version, I felt that most of the original comments I made still stand. Unfortunately, the serious comments. While the idea and work are interesting, academic rigor remains required for presentation. The literature review and theoretical framework remain weak. They do not present a thorough examination of the various bodies of literature available to you. Some of the readings I suggested (none of which are my own work), I would have like to see built into the presentation of your literature and discussion. The nature of the existing course and the ways in which it was redesigned in light of the literature remains unclear too. The presentation remains very general in nature.

Thank you.

Author Response

We sincerely thank you again for taking the time to review our paper. Your efforts and feedback mean a lot to us as they assist in enhancing the quality of our work. We confirm that all the comments and feedback provided have been addressed in our revised manuscript. We have also incorporated ideas from three suggested articles.

We admit that some ideas we shared could be presented in different ways. So, we have addressed the fundamental issues, while our styles may differ in some parts of the paper.
